# Ion mobility-tandem mass spectrometry of mucin-type *O*-glycans

**Leïla Bechtella** [1,2], **Jin Chunsheng** [3], **Kerstin Fentker** [1,4], **Güney R. Ertürk**[1], **Marc Safferthal** [1,2], **Łukasz Polewski** [1,2], **Michael Götze** [1,2], **Simon Y. Graeber** [5,6,7], **Gaël M. Vos**[1,2], **Weston B. Struwe** [8,9], **Marcus A. Mall** [5,6,7], **Philipp Mertins** [4,10], **Niclas G. Karlsson**[3,11] & **Kevin Pagel** [1,2] ✉

The dense *O*-glycosylation of mucins plays an important role in the defensive properties of the mucus hydrogel. Aberrant glycosylation is often correlated with inflammation and pathology such as COPD, cancer, and Crohn's disease. The inherent complexity of glycans and the diversity in the *O*-core structure constitute fundamental challenges for the analysis of mucin-type *O*-glycans. Due to coexistence of multiple isomers, multidimensional workflows such as LC-MS are required. To separate the highly polar carbohydrates, porous graphitized carbon is often used as a stationary phase. However, LC-MS workflows are time-consuming and lack reproducibility. Here we present a rapid alternative for separating and identifying *O*-glycans released from mucins based on trapped ion mobility mass spectrometry. Compared to established LC-MS, the acquisition time is reduced from an hour to two minutes. To test the validity, the developed workflow was applied to sputum samples from cystic fibrosis patients to map *O*-glycosylation features associated with disease.

Mucins are the main component of mucus, a barrier-like hydrogel that is especially prominent in the enteric and pulmonary epithelia[1]. They are large proteins that are heavily decorated with *O*-glycans, which constitute up to 80% of their molecular weight. Their glycosylation profile is highly heterogeneous and significantly altered in diseases such as cystic fibrosis (CF)[2], chronic obstructive pulmonary diseases (COPD), or cancer[3], suggesting a role in disease progression and strong diagnostic potential. However, very little is known about the glycosylation molecular structures and the consequences on the glycan-dependent macromolecular interactions of the mucus. A fundamental challenge for the analysis of mucin-type *O*-glycans is the high complexity of glycan structures; the constituting monosaccharides are often isomeric (e.g. GalNAc vs. GlcNAc), the building blocks can be linked in a variety of positions (e.g. 1-3 vs. 1-6), and the glycosidic bond can be oriented in different configurations (α vs. β anomers). This leads to highly branched and isomeric oligosaccharides, which often coexist but potentially have varying functional properties. These isomers cannot be easily distinguished by mass spectrometry alone[4] and therefore orthogonal techniques are usually required for their identification. A second fundamental challenge arises from *O*-link core diversity[5]. In contrast to *N*-glycans, which share a conserved Man3GlcNAc2 core, eight different core structures are found for

[1]Institute of Chemistry and Biochemistry, Freie Universität Berlin, Altensteinstraße 23A, 14195 Berlin, Germany. [2]Fritz Haber Institute of the Max Planck Society, Faradayweg 4-6, 14195 Berlin, Germany. [3]Department of Medical Biochemistry and Cell Biology, Institute of Biomedicine, Sahlgrenska Academy, University of Gothenburg, Gothenburg, Sweden. [4]Max Delbrück Center for Molecular Medicine, Robert-Rössle-Str. 10, 13125 Berlin, Germany. [5]Department of Pediatric Respiratory Medicine, Immunology and Critical Care Medicine and Cystic Fibrosis Center, Charité – Universitätsmedizin Berlin, corporate member of Freie Universität Berlin and Humboldt-Universität zu Berlin, Berlin, Germany. [6]German Center for Lung Research (DZL), associated partner site, Berlin, Germany. [7]Berlin Institute of Health at Charité – Universitätsmedizin Berlin, Berlin, Germany. [8]Kavli Institute for Nanoscience Discovery, University of Oxford, Oxford OX1 3QU, UK. [9]Department of Biochemistry, University of Oxford, Oxford OX1 3QU, UK. [10]Berlin Institute of Health, 10178 Berlin, Germany. [11]Department of Life Sciences and Health, Faculty of Health Sciences, Oslo Metropolitan University, Oslo, Norway. ✉e-mail: kevin.pagel@fu-berlin.de

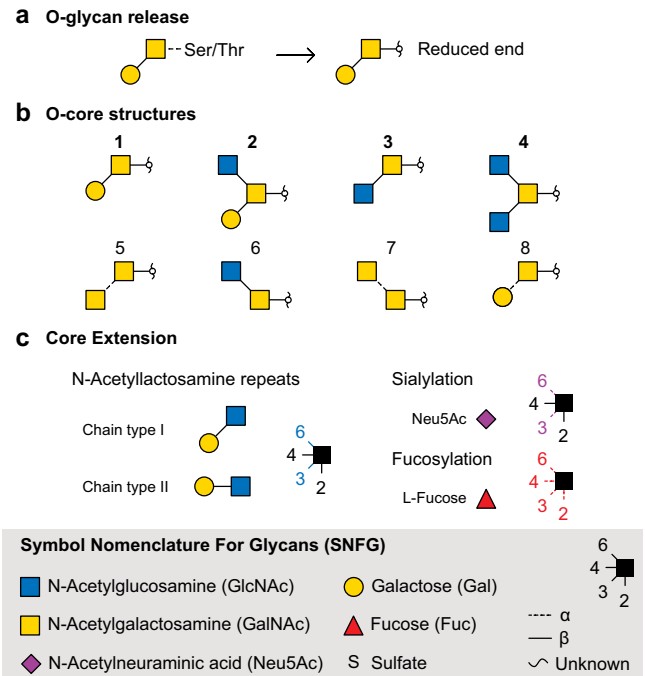

**a O-glycan release**

**b O-core structures**

**c Core Extension**

N-Acetyllactosamine repeats

Chain type I

Chain type II

Sialylation

Neu5Ac

Fucosylation

L-Fucose

**Symbol Nomenclature For Glycans (SNFG)**

- N-Acetylglucosamine (GlcNAc)
- N-Acetylgalactosamine (GalNAc)
- N-Acetylneuraminic acid (Neu5Ac)
- Galactose (Gal)
- Fucose (Fuc)
- S Sulfate
- ---- α
- — β
- ⌢ Unknown

**Fig. 1 | Typical *O*-glycan structures shown using the symbol nomenclature for glycans (SNFG)**[59]. **a** Representation of the *O*-glycosidic linkage to Ser/Thr and the alditol form resulting from β-elimination. **b** The most abundant *O*-glycan core structures 1–4 and the four less common cores 5–8. **c** Typical extensions of glycan cores are the attachment of *N*-acetyllactosamine repeats, sialylation and fucosylation.

*O*-glycans (Fig. 1). Mucin-type *O*-glycans are extended from a common GalNAc bound to the glycoprotein through the hydroxy group of a Ser or Thr side chain. As there is no universal enzyme able to cleave all *O*-glycan core structures collectively, chemical methods such as reductive β-elimination are usually employed. The released *O*-glycans are then converted to alditols, to prevent subsequent peeling reaction leading to the loss of the GalNAc at the reducing end[6,7], and dilution of the signal over the GalNAc α and β anomers[8].

Typically, liquid chromatography (LC) coupled to tandem mass spectrometry (MS/MS) is employed for the analysis of mucin-type *O*-glycans[8,9]. A stationary phase of porous graphitised carbon (PGC) is commonly used for the separation of non-derivatized *O*-glycans. The polar carbohydrates are well retained on the polarisable PGC phase[10], while its planar surface adds a 3-dimensional separation[11]. Although recognised for its ability to separate isomers, PGC tends to lack reproducibility[12,13]. To maintain its separation properties, regular regeneration steps using acids or methanol are required throughout the analysis sequence[12,13]. LC retention times are used as an additional level of identification for putative assignment of glycan structures, but these often vary significantly. Deviations in retention times are seemingly dependent on both the type of instrument as well as the employed experimental conditions, making inter-laboratory comparison difficult.

Following their separation, glycan isomers can be identified by tandem MS, typically via collision-induced dissociation (CID) of negatively charged ions. CID of deprotonated glycans leads to cross-ring fragments, which are diagnostic to the regiochemistry and core structure[14–16]. Although highly informative, glycan MS/MS spectra are characteristically complex and difficult to interpret. An orthogonal structural technique to distinguish and identify isomeric structures with higher confidence is therefore highly desirable.

Ion mobility spectrometry (IMS) has recently emerged as a powerful technique to separate isomeric glycans in the gas phase[17–19]. In traditional IMS devices, ions are propelled by an electric field through a mobility cell filled with an inert gas such as He or $N_2$. Low-energy collisions with the gas facilitate a temporal separation of ions according to their size, shape, and charge. The collision cross section (CCS) retrieved from mobility experiments is a molecular property and corresponds to the rotationally averaged area of the ion that collides with the drift gas[20]. Depending on the type of instrument, CCSs can be measured directly or estimated after calibration. Unlike _PGC_LC retention times, the CCS of an ion is instrument independent and can in principle be calculated theoretically. Furthermore, deprotonated glycans can be separated more efficiently in IMS than their protonated or sodiated forms[18], while being less prone to rearrangement reactions such as fucose migration[21–23].

Recently, the potential of ion mobility spectrometry for the separation of *O*-glycan structures was tested and compared to _PGC_LC separation[24]. This work clearly demonstrated the potential for IMS to aid in the identification of otherwise challenging to assign *O*-glycan isomers through their CCS. However, the resolution achieved with the utilised travelling wave ion mobility spectrometer (TWIMS) was not sufficient to separate the majority of intact glycan ions and estimate their CCS. Instead, mobilities of diagnostic fragments had to be used to identify the parent structures.

The resolving power of traditional IMS techniques, such as drift tube IMS and TWIMS, is mostly governed by the dimensions of the drift cell. Recent advances in instrumentation address this issue either by performing cyclic IMS experiments in a closed-loop system[25], or by using distinct cell geometries and electric fields. In trapped ion mobility spectrometry (TIMS)[26], ions are driven by the drift gas through the cell and trapped by an electric field going in opposite direction. The progressive decrease of the electric field separates the ions according to their mobility. By manipulating the ramp of the electric field and the gas flow, the resolving power can be increased without altering the mobility cell.

Here, we present a TIMS-based method to separate and comprehensively characterise glycan isomers from complex biological samples. The complete analytical workflow takes only a few minutes, which makes it an attractive alternative to the rather slow LC-MS method and importantly improves IMS resolution as compared to aforementioned studies. Additionally, the measured CCSs enable a database-driven high-throughput *O*-glycan analysis.

## Results

### TIMS separation of *O*-glycan isomers

To validate the adequate resolving power of TIMS for natural mucin-type *O*-glycan analysis, the *O*-glycome of commercially available porcine gastric mucins (PGM) was investigated. PGM was selected for its highly heterogeneous but well-characterised *O*-glycosylation[24]. The released glycan alditols were ionised through direct infusion and separated in the TIMS cell. Multiple IMS scans of 1 s ramp time were accumulated over two minutes to reach highest signal-to-noise ratio. In just two minutes, a detailed TIM-MS *O*-glycoprofile of PGM comparable to that of a 40 min _PGC_LC-MS gradient was acquired (Fig. 2).

Most of the observed ions are singly deprotonated; doubly deprotonated species are only detected for glycans above 1200 Da. Figure 2a shows the 2D plot of the released PGM *O*-glycans with *m/z* on one the y-axis and $1/K_0$ on the x-axis. The highlighted region shows the trendlines of singly and doubly charged ions. The total mobilogram can be annotated in the same way as an LC chromatogram, using the CCS instead of the retention time in min (Fig. 2b). The general composition of the glycans was retrieved from the *m/z* values. Complete *O*-glycan structures were assigned from MS/MS spectra (for complete assignment see Supplementary Data 1). Forty-nine structures were identified in PGM, with in average two isomers per composition. Overall, the TIM-MS accumulated spectrum matches the _PGC_LC-MS data well (Fig. 2c).

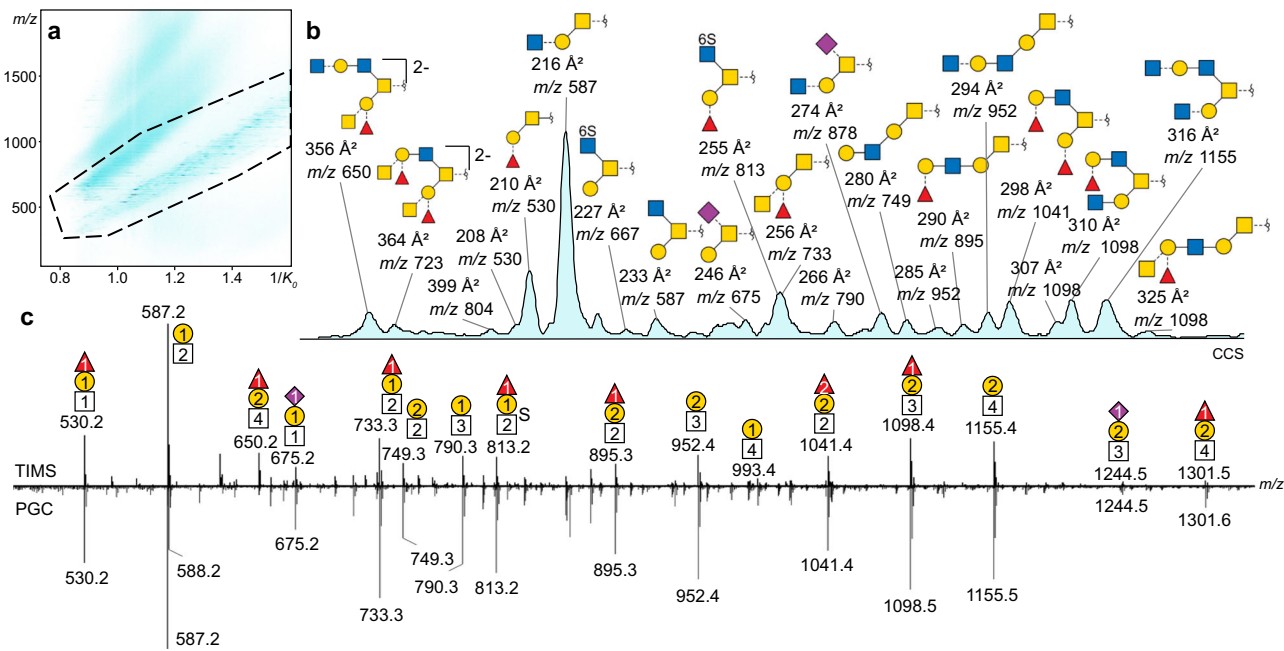

**Fig. 2 | TIM-MS of deprotonated *O*-glycans from porcine gastric mucins. a** TIM-MS *m/z* and $1/K_0$ heatmap with highlighted singly and doubly deprotonated *O*-glycan ions (dashed line). **b** Extracted ion mobilogram with collision cross section (CCS) and *m/z* of the most abundant species. **c** MS spectra accumulated over the full ramp time/gradient in TIMS (2 min) and ₚGCLC (40 min). Glycan compositions and structures are shown using the SNFG nomenclature[59]. The number of each residue is noted inside the symbol. All assigned structures, their CCSs, and their corresponding fragmentation spectra are given in the Supplementary Data 1.

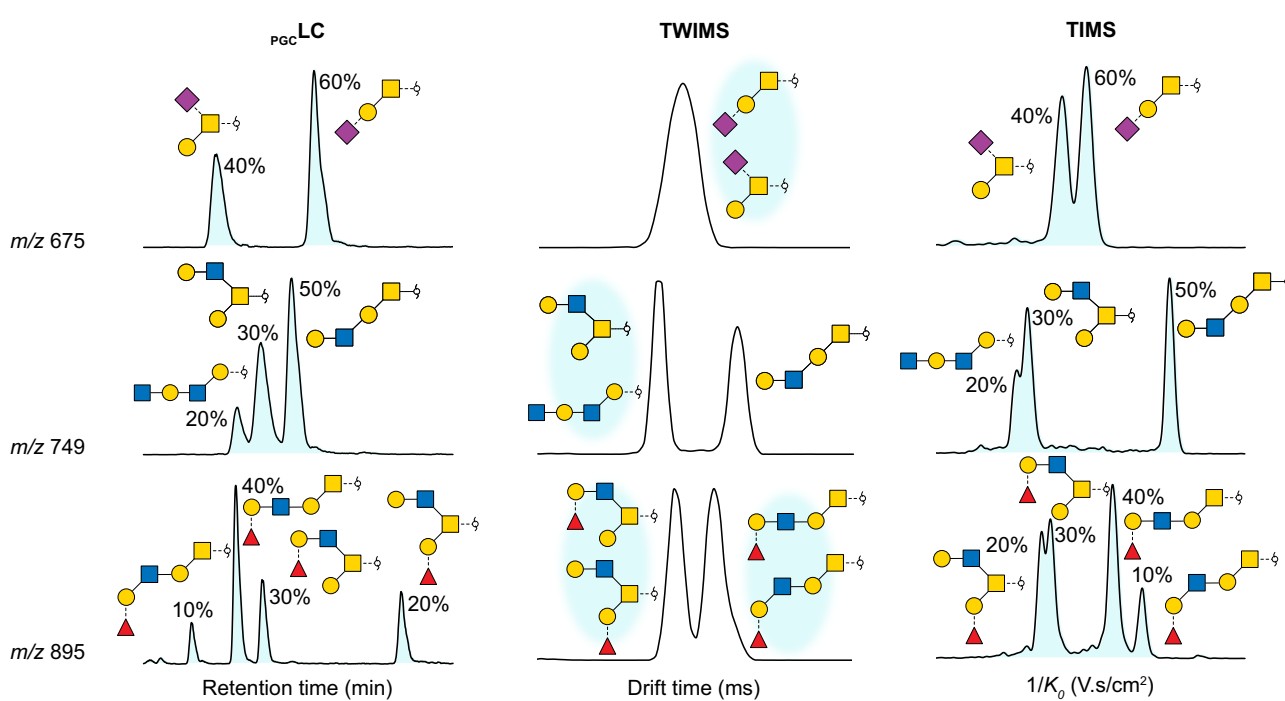

**Fig. 3 | Efficiency of isomeric separation for a selection of three mucin-type *O*-glycans.** The extracted ion chromatograms (ₚGCLC) and mobilograms (TWIMS and TIMS) of deprotonated ions at *m/z* 675, 749 and 895 are shown. The relative area of each isomer is indicated in percentage.

Separation of glycan isomers is key for the structural characterisation of the *O*-glycome. ₚGCLC is a powerful method for *O*-glycan separation. However, it is also time-consuming and prone to error as retention time shifts can induce false putative assignments. TWIMS has previously also been tested for *O*-glycans analysis, albeit with limited resolving power[24]. In order to assess the utility of the method presented here, the separation of selected deprotonated isomeric structures using ₚGCLC, TWIMS and TIMS are compared (Fig. 3). The relative peak area for each isomer is annotated in percentage of the sum area of isomeric structures at same *m/z*. While all *O*-glycan isomers are separated using ₚGCLC, TWIMS clearly lacks the resolution to distinguish all isomers. TIMS on the other hand has a significantly improved resolution and sufficiently separates all structures that can be distinguished by ₚGCLC. In addition, the relative peak areas of

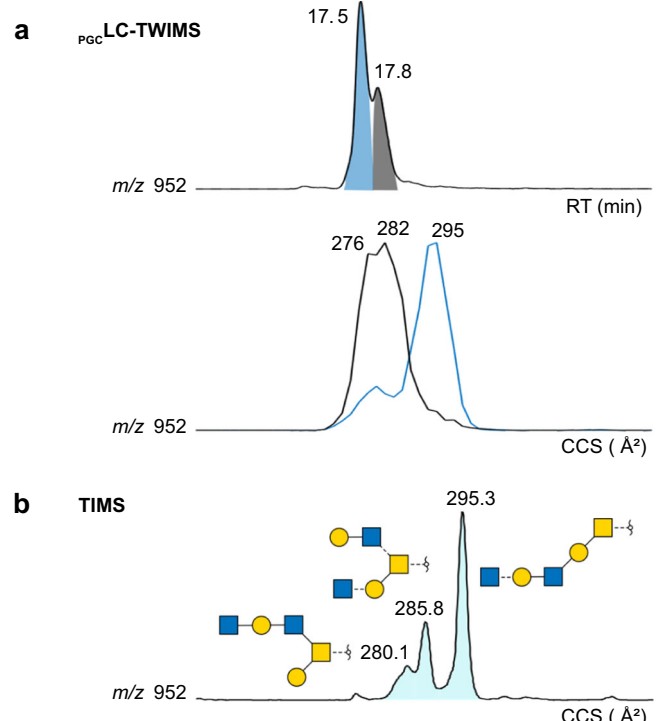

**Fig. 4 | Separation of neutral isomeric *O*-glycans at *m/z* 952 using PGCLC coupled to TWIMS or stand-alone TIMS. a** Extracted ion chromatogram of *m/z* 952 and corresponding ion mobilograms (TWIMS) for each PGCLC peak. **b** Extracted ion mobilogram (TIMS) of *m/z* 952.

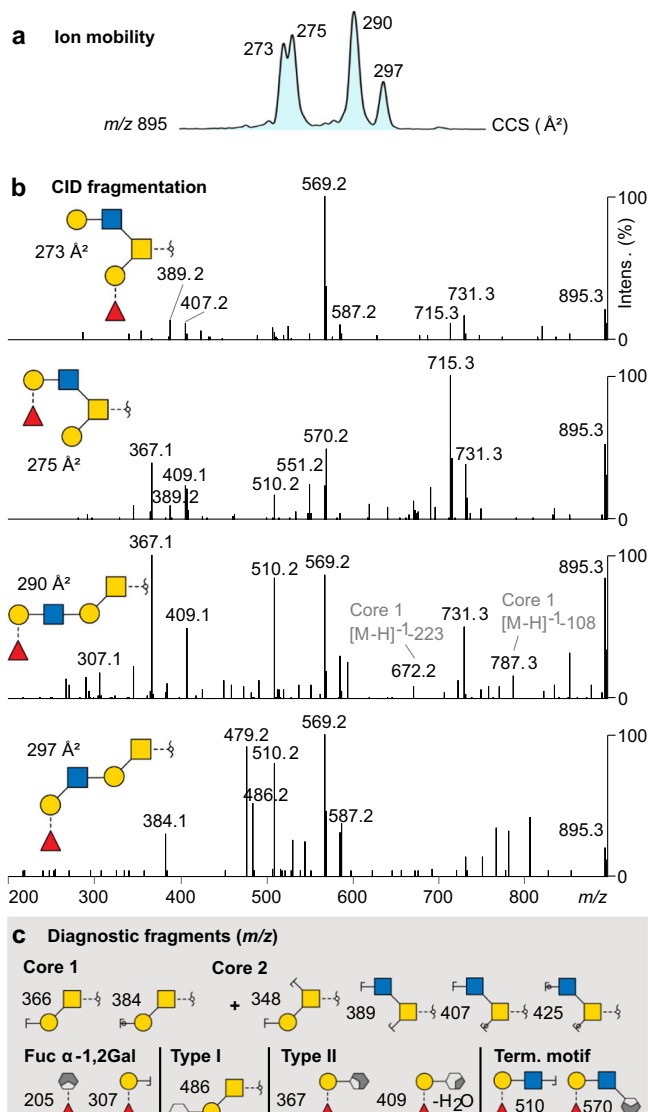

**Fig. 5 | TIM-MS/MS structural assignment based on characteristic alditols fragmentation. a** Extracted TIM ion mobilogram of *m/z* 895. **b** Accumulated CID fragmentation spectra for each mobility peak of *m/z* 895 and corresponding identified structures. **c** Main characteristic fragments that aided the assignment of the four isomeric structures.

selected structures do not differ drastically between PGCLC and TIMS, which indicates that intensities can be compared for semi-quantification.

Some neutral structures are challenging to separate using PGCLC and TWIMS (Fig. 4a). For both compositions, the direct hyphenation of PGCLC and TWIMS reveals the existence of several isomers, but overlapping peaks prevent further characterisation. Due to the improved mobility resolution, TIMS can address this issue. All isomers can be resolved sufficiently (Fig. 4b), and the resulting fragmentation spectra allow the identification of their structures (SI).

## TIM-MS/MS for glycans structural assignment

In a TIMS experiment, isomers are first separated in the mobility cell and subsequently analysed by MS/MS. The TIM cell is placed directly after the ionisation source, the mobility-separated ions are directed to the quadrupole for precursor ion selection, then to the collision cell for fragmentation and finally to the TOF for mass detection. As a result, the TIM-MS/MS workflow is conceptually comparable to that of LC-MS/MS, where an individual fragmentation spectrum can be obtained for each LC-separated species. This is shown in Fig. 5 where four isomers of *m/z* 895 are separated by TIMS and subsequently fragmented by MS/MS. The fragmentation of negatively charged glycans yields highly informative cross-ring fragments, from which the regiochemistry and the glycan core structure can be deduced[15,16,27–29]. Here, diagnostic fragments indicate a core 2 for the structures of CCSs 273 Å$^2$ and 275 Å$^2$ and a core 1 for the structures of CCSs 290 Å$^2$ and 297 Å$^2$. An α−1,2 fucosylated Gal is found in all four isomers. A blood group H and a type II chain were attributed to the structures of CCS 275 Å$^2$ and 290 Å$^2$. The structure of CCS 297 Å$^2$ is a core 1 extended with a blood group H and a type I chain. *O*-glycan structures in PGM were systematically assigned and are all reported in the Supplementary Data 2. The results match well with previously reported structures[24].

## CCS-based structural characterisation

Retention times are governed by multiple factors such as the physiochemical properties of the analyte and stationary phase. As a result, they often lack reproducibility. CCSs on the other hand are molecular properties that are directly linked to the structure of the ions in the gas phase. This makes CCS values particularly interesting for characterising isomers, where they can provide important information on the branching and regiochemistry of the investigated molecules.

Isomer separation by ion mobility can reveal minute structural differences. The ions at *m/z* 895 discussed above are a good example. The two structures of CCS 290 Å$^2$ and 297 Å$^2$ differ only by the presence of a β−1,4 or β−1,3 glycosidic bond, corresponding to type I and type II linkages, respectively (Fig. 5). This linkage detail is often challenging to identify by MS/MS alone and depends on the presence of diagnostic, but often low-abundant, cross-ring fragments. The type I structure yields a higher CCS than type II, which can be explained by the presence of the β−1,3 linkage inducing a more open structure. These observations can help assign similar structures more generally.

Aside from qualitative use, CCSs can be utilised for de novo annotations. CCSs are universally comparable when measured under controlled conditions[30]. As a result, they can be readily implemented in online databases and used independently of the instrument and laboratory[31]. Direct CCS measurement can be performed on drift tube IMS instruments; other IMS approaches require calibration with calibrants of similar physico-chemical properties as the analyte to estimate CCS values. The CCSs of PGM $O$-glycans reported here were determined using two ion mobility spectrometers, TIMS and TWIMS, which use considerably different technologies to achieve mobility separation. The standard deviation between $^{TIMS}CCS_{N2}$ and $^{TWIMS}CCS_{N2}$ is consistently below 1% over the whole CCS range, which demonstrates the general validity of the approach. All CCS values and the corresponding STDs between the two instruments are given in the Supplementary Data 2. The processed data is available on the online database UniCarb-DR, with CCS values indicated in the place of retention times. The accession codes are provided in the Data Availability section.

### Clinical sputum $O$-glycan profiling

As a proof-of-principle, the TIM-MS/MS method was applied to two clinical samples of healthy and cystic fibrosis (CF) sputum. Patients with CF produce a substantial amount of sputum as the disease is associated with a hypersecretion of mucus. For this reason, the glycosylation of secreted mucins in CF sputum has been thoroughly investigated. The stage of the muco-obstructive lung disease induces drastic variations in the mucus $O$-glycan profile, as previously reported[2,32–34]. Here, healthy and CF sputum samples are used to verify if the TIM-MS/MS method can discriminate the two clinical samples and highlight prominent glycosylation alterations in CF sputum. First, the relevance of the samples to the disease state was assessed using proteomics. Of the 2283 quantified proteins, 1375 have a significantly different abundance (adj. $p$-value < 0.05). The complete protein identification output is reported in Supplementary Data 3. The main contributors to glycosylation in the sputum with a molecular weight above 500 kDa are mucins. Therefore, we compared the log2 intensities of the four quantified high molecular weight mucins (Fig. 6). MUC5B is the most abundant mucin in both samples and therefore likely contributes the most to the glycosylation profiles. In the healthy sample, the contribution is higher than in the CF sample. MUC5AC is the second most abundant mucin and higher abundant in the CF sputum. MUC2 has only been quantified in the CF sample and MUC1 contributes the least to the glycosylation because of its low abundance. Overall, the data shows that the sputum samples contain MUC4, MUC1, MUC16, MUC7, MUC5AC and MUC5B. The increased level of MUC5AC and decreased level of MUC5B in CF has been reported previously[35,36]. The proteomics data therefore confirm that the chosen samples follow the general trends observed in CF samples. As size exclusion chromatography with a mass cut-off of 600 kDa was used to collect the mucin fraction before performing glycomics, we assume that the sputum samples used for glycomics contain mostly MUC5AC and MUC5B.

TIM-MS/MS analysis reveals strong differences in the $O$-glycosylation profile of healthy and CF sputum (Fig. 7a, for complete assignment, see Supplementary Data 1 and 2). In general, the glycans are much simpler in the CF sample and decorated to a higher degree with sialic acid. The diversity in glycan structures is reduced in the CF sample, for which the mobilogram area seem to be concentrated on a few peaks of high intensity. This agrees well with previous studies revealing a generally reduced complexity in CF glycosylation[2].

To compare the relative abundance of core structures and specific glycan features such as sialylation, fucosylation and sulfation, the mobility peak areas have been integrated and combined into groups (Fig. 7b, c). The CF sample presents less core 3 and core 4 compared to the healthy one, reflecting a significant loss of diversity

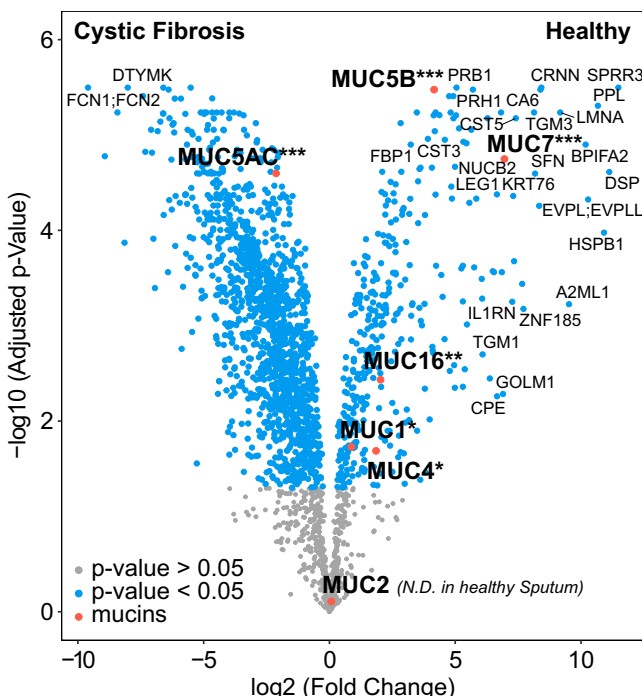

| Mucin | MW (kDa) | Log2(LFQ) | | |
| | | CF | Healthy | Adj. p-Value |
|---|---|---|---|---|
| MUC5AC | ~2200; >500 | 29.8 | 27.7 | 0.000 *** |
| MUC2 | ~550 or higher | 24.1 | N.D. | |
| MUC1 | 200-500 | 25.7 | 26.6 | 0.194 ns |
| MUC4 | 320 | 24.0 | 25.9 | 0.252 ns |
| MUC16 | >250 | 25.1 | 27.2 | 0.033 * |
| MUC7 | 150-200 | 24.2 | 31.1 | 0.043 * |
| MUC5B | 1000-2000 | 30.4 | 34.6 | 0.000 *** |

**Fig. 6 | Comparison of protein abundances between healthy and cystic fibrosis sputum.** Proteins of significantly different intensities (adjusted $p$-value < 0.05, moderated $t$-test) are highlighted in blue. Additionally, mucin proteins are highlighted in red. The table indicates mean Log2 label-free quantification (LFQ) intensities and significance for each mucin calculated using a two-sided $t$-test. A Benjamini–Hochberg method was used to adjust for multiple comparisons. *** adj. $p$ < 0.001, ** adj. $p$ < 0.01, * adj. $p$ < 0.05. The molecular weight is indicated for each mucin[60–70]. Source data are provided as a Source Data file.

in the CF core structures. Sialylated glycans are three times more abundant in the CF sputum, while fucosylation and sulfation are decreased. Similar trends were reported in the past using $_{PGC}$LC-MS on purified MUC5B and MUC5AC[32,37]. Specifically, the high level of sialylation in CF sputum and inflammatory state is consensus of various studies[32,34,37,38].

Closer inspection of the relative abundance of isomeric structures provides a deeper level of information (Fig. 8). Even when all isomeric structures are present in the sample, the peak ratios can identify differences in their abundance. In case of the trisaccharides at $m/z$ 587, these changes are notable, with a complete inversion in abundance. A simple qualitative analysis of the samples would not reveal such differences.

The TIM-MS method reported here can provide $O$-glycan profiles from clinical samples in as little as two minutes. In addition, the CCSs of all identified $O$-glycans were determined and can be used as references

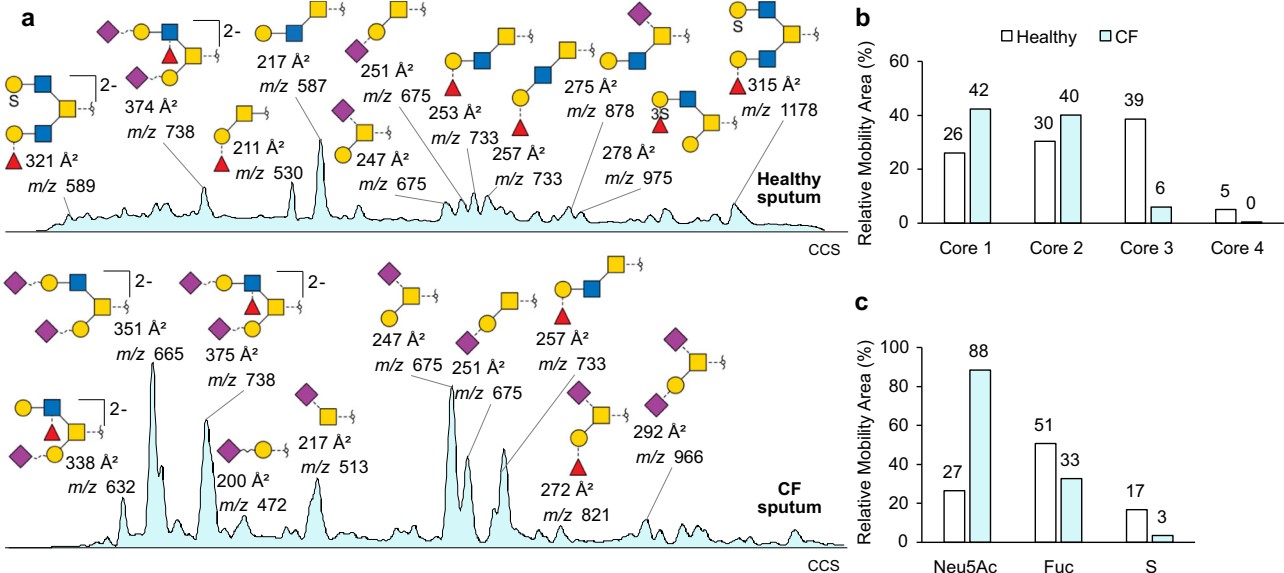

**Fig. 7 | *O*-glycosylation profile of clinical sputum by TIM-MS/MS. a** Glycosylation profiles of healthy sputum (top) and cystic fibrosis (CF) sputum (bottom) obtained by TIM-MS/MS. The content in **b** core structures and **c** glycosylation features (sialylation, Neu5Ac; fucosylation, Fuc; and sulfation, S) are shown in relative mobility areas. The detailed fragmentation data is reported in Supplementary Data 1 and all assigned structures with their corresponding CCS values and mobility peak area are reported in the Supplementary Data 2.

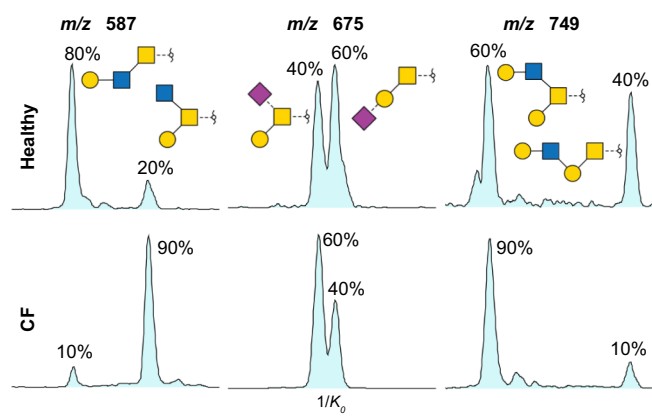

**Fig. 8 | Abundance of isomeric structures in healthy and cystic fibrosis (CF) samples.** Extracted ion mobilograms of *m/z* 587, 675 and 749 (from left to right) in healthy (top) and CF (bottom) sputum. The relative area of each isomer is indicated in percentage.

for future assignment. This combination of short analysis time and diagnostic CCSs enables a high-throughput analysis of *O*-glycosylation features in clinical samples.

## Discussion

For the analysis of complex samples, it is crucial to separate their constituents, especially when multiple isomers coexist as in the case of glycans. Liquid chromatography can profile *O*-glycans in approximately one hour. The work presented here shows that ion mobility spectrometry is much faster and provides a qualitatively similar overview in just a couple of minutes, which makes it a powerful alternative to classical LC-MS workflows.

Until now, the use of ion mobility in the omics fields is largely based on implementation into existing LC-MS workflows as a third dimension. Here, we provide a method that utilises IMS as an independent, stand-alone separation technique for the analysis of complex, highly heterogeneous samples. We show that the stand-alone

TIMS technology with the well-tuned mobility parameters leads to mobilograms that in terms of resolution and informational content resemble LC chromatograms. In addition to the rapid separation, IMS provides direct structural information in the form of collision cross sections, which can be utilised for structural assignment in complex omics analyses.

The structural complexity of glycans is still the major bottleneck for their analysis. Recent advancements in ion mobility technology highlighted the potential of the method to be used in LC-based glycomics analyses[24,39–44]. Fundamental studies recently showed that IMS can be theoretically described using a plate-height model, similarly to other separation techniques such as capillary zone electrophoresis or LC[45,46]. Dedicated bioinformatics tools for deconvolution and data treatment further strengthen the utility of ion mobility in real-world applications[31,47,48]. Finally, also the often tedious fragmentation-based assignment of *O*-glycans structures is currently on the verge of automation, for example, via deep-learning approaches[49]. When combined, these advancements have the potential to transform glycomics from a highly specialized technology into a broader, high-throughput application.

## Methods

### Ethical statement

This research complies with all relevant ethical regulations. The collection of sputum samples was approved by the ethics committee of Charité - Universitätsmedizin Berlin (EA2/016/18). Written informed consent was obtained from all participants.

### Sputum collection

Two sputum samples were used in this study: one of a healthy donor and one of a cystic fibrosis patient, and consent has been obtained for sharing of individual-level data. Samples were matched for sex and gender, both female, and age at the time of collection, was between 28 and 33. Sex and gender were therefore not considered in the study design. The healthy sample was collected after induction with inhaled hypertonic saline (NaCl 6%) and the CF sample was collected after spontaneous expectoration. Samples were stored at −80 °C until analysis.

## O-glycan sample preparation

Porcine gastric mucin (PGM) was obtained from Sigma-Aldrich (U.S.A.). The clinical mucus samples were first heat-inactivated. In the same step, samples were reduced and alkylated, and detergent was added to the mixture to disrupt the mucus hydrogel. Briefly, an equal volume of buffer containing 4% SDS, 100 mM Tris-HCl pH 8, 1 mM EDTA, 150 mM NaCl, 20 mM DTT, 80 mM CAA was added to the mucus. The mixture was heated at 95 °C for 10 min. After cooling down, the samples were incubated with 25 U of benzonase for 15 min at room temperature. The mucins fraction of the samples was purified by size exclusion chromatography (SEC) using an AZURA Multi Method FPLC System (Knauer, Germany). Superdex 200 (prep grade, Cytiva, USA) was used to pack a column of 37 mL bed volume. The separation was performed on 200 μL of inactivated sample, at a flow rate of 0.75 mL/min in 100 mM ammonium acetate. Absorbance was recorded at 214 nm. The mucin fraction was collected in the void volume, lyophilised, and resuspended in 200 μL H$_2$O. O-glycans from PGM (2 mg/mL in H$_2$O) and from the collected sputum mucins were released by reductive β-elimination[8]. Samples (160 μL) were incubated with 20 μL 5 M NaBH$_4$ and 20 μL 0.5 mM NaOH at 50 °C for 16 h. The reaction was quenched with 20 μL acetic acid and desalted on 400 mg Dowex 50WX8 cation exchange beads (Sigma-Aldrich, USA). The resin was first washed three times with methanol (MeOH), and conditioned with 1 mL HCl 1 M, 1 mL MeOH and 1 mL H$_2$O. Glycans were loaded on the resin and eluted with two times 500 μL H$_2$O. O-glycans were then enriched on a 50 mg Hypercarb SPE cartridge (Thermo Fischer). The cartridge was conditioned with 1 mL ACN 0.1% TFA, and 1 mL H$_2$O 0.1% TFA. Glycans were loaded on the cartridge and washed three times with 1 mL H$_2$O 0.1% TFA. Glycans were eluted with four times 100 μL of 50% ACN 0.1% TFA and dried in a SpeedVac.

## O-glycan analysis by PGCLC-MSⁿ and TWIMS

PGCLC-MS, PGCLC-MS/MS and PGCLC-TWIMS were performed using a SYNAPT G2-Si spectrometer (Waters, Manchester, U.K.) equipped with an Acquity UPLC system. O-glycan alditols were separated at room temperature using a 100 × 2.1 mm I.D. PGC column of 5 μm particle size (Hypercarb, Thermo Scientific, U.S.A.). The released O-glycans were dissolved in 40 μL H$_2$O and 5 μL were injected. Glycans were eluted using a linear gradient from 0 to 40% acetonitrile (ACN) in 10 mM NH$_4$HCO$_3$ over 40 min at a flow rate of 150 μL/min. Glycan alditols were ionised in ESI negative ion mode with a capillary voltage of 2.8 kV, and the source temperature was 150 °C. The mass range was set to m/z 50-3500. Fragmentation was performed using ramps of collision energy dependent on the m/z ratio of the precursor ions, in the range of 15 to 120 eV. O-glycan mobilities were recorded after LC separation in a separate acquisition, using an IMS wave velocity of 450 m/s, and an IMS wave height of 40 V. The data was acquired and processed using the MassLynx software (version 2.0.7, Waters). To obtain relative peak areas, the chromatograms were deconvoluted and integrated with MZmine 2[50]. O-glycans structures were manually assigned from the MS/MS spectra, based on their diagnostic fragment ions, using GlycoWorkBench 2.1[51].

## O-glycan analysis by TIM-MSⁿ

TIM-MS and TIM-MS/MS were performed on a timsTOF Pro spectrometer (Bruker, Bremen, Germany), equipped with an in-house 3D-printed offline nano-ESI source, whose design was recently published[52]. Released glycans were dissolved in 50 mM ammonium acetate in H$_2$O:MeOH (1:1). For each measurement, 5 μL of sample were introduced in a Pt/Pd-coated glass capillary emitter prepared in-house and ionised in negative ion mode nano-ESI. The instrument parameters were set as follows: capillary voltage 1.2 kV, end plate offset −0.5 kV, dry source temperature 150 °C, D1 = 20 V, D2 = 30 V, D3 = −100 V, D4 = −450 V, D5 = 60 V, and D6 = −46 V. The TIMS separation was performed in

nitrogen (N$_2$). Ions reversed mobility $1/K_O$ was scanned between 0.6 V·s/cm$^2$ and 1.6 V·s/cm$^2$. The accumulation time and the ramp time were set to 20 ms and 1000 ms, respectively. The transfer time was fixed to 2 ms. Ions were selected in the quadrupole after mobility separation and fragmented in the collision cell, with collision energy varying from 25 to 120 eV. The data was acquired with Compass otofControl (version 6.2, Bruker). The mobilograms were processed and integrated using DataAnalysis (version 4.0, Bruker). O-glycans structures were manually assigned using GlycoWorkBench 2.1[51].

## Mobility calibration and data treatment

TWIMS arrival time distributions were first fitted to a Gaussian distribution with Origin 95E before estimating ᵀᵂᴵᴹˢCCSs. TWIMS calibration was performed based on a described protocol[53], using a dextran ladder 1 K (Sigma-Aldrich) of known drift tube CCS in N$_2$, ᴰᵀCCS$_{N2}$[54,55]. Briefly, a logarithmic calibration curve was plotted using the corrected drift time $t_D'$ (in ms) and corrected CCS' (in Å$^2$), using the following equations:

$$t_D' = t_D - C\frac{\sqrt{m/z}}{1000}, \tag{1}$$

where $t_D$ is the experimental drift time and $C$ is the EDC Delay Coefficient (1.41).

$$CCS' = CCS \times \frac{\sqrt{\mu}}{z}, \tag{2}$$

where $\mu$ is the reduced mass.

The fitted drift time $t_D$ was corrected by subtracting the travel time from the end of the mobility cell to the entrance of the mass spectrometer in (1). The reference CCS value was corrected by the reduced mass and charge in (2).

As TIMS experiments result in the temporal release of the ions, dependent on the electric field gradient, and not in a drift time, a calibration is needed to convert the ion transit time $t_t$ in the ion mobility cell into a reversed mobility $1/K_O$[26]. This was achieved through the otofControl software (Bruker), using the ESI-Low-concentration tuning mix from Agilent. TIMS $1/K_O$ values of the dextran ladder 1 K were then recorded and plotted against their corrected CCS' (in Å$^2$), using Eq. (2). This external correction of the TIMS mobilities led to better results in terms of inter-instruments reproducibility of the CCS$_{N2}$.

All ᵀᵂᴵᴹˢCCS$_{N2}$ and ᵀᴵᴹˢCCS$_{N2}$ values are listed in the Supplementary Data 2, which presents averages of two or three replicates acquired in independent measurements on each instrument.

## Proteomics of clinical samples

Sputum samples were inactivated in SDS buffer (2% SDS, 50 mM Tris-HCl pH 8, 0.5 mM EDTA) at 95 °C for 10 min. After measuring the protein concentration using BCA assay, 100 μg protein were reduced and alkylated in SDS buffer with 20 mM DTT, 40 mM CAA and 75 mM NaCl at 95 °C for 10 min. Subsequently, the mixture was incubated with 25 U benzonase for 15 min at room temperature and proteins were extracted using an SP3 protocol described for proteomics[56]. Sera-Mag SpeedBeads (GE Healthcare, cat. no. 45152105050250 and 65152105050250) were prepared in a 1:1 mix at 50 mg/mL. Proteins were precipitated in 70% ACN and incubated with 5 μL of the beads mix, at 1 μg protein:2.5 μg beads, for 20 minutes under shaking (800 rpm). The beads were washed twice with 70% ethanol and once with 100% ACN on a magnetic rack. After removal of the ACN, beads were resuspended in 50 mM ABC buffer and incubated with PNGase F for 1 h at 37 °C. Trypsin and LysC were added at 1:50 (enzyme:substrate) for overnight digestion. The supernatant was collected and

desalted on C18 material. The analysis was performed using an EASY-nLC 1200 System coupled to an Orbitrap Exploris 480 mass spectrometer (Thermo Fisher Scientific). Peptides (2 μL at 1 μg/μL) were injected two and three times for healthy and CF samples, respectively. Peptides were separated at 45 °C using a 20 cm reverse-phase column (Fused Silica Capillary I.D. 75 μm, CM Scientific), packed in-house with 1.9 μm C18-reprosil beads (r119.aq.0003, Dr. Maisch). Eluents A (3% ACN, 0.1% FA) and B (90% ACN, 0.1% FA) were used. Peptides were eluted at a flow rate of 250 nL/min, applying the following 200 min gradient: 2% to 3% B over 5 min, then from 3% to 20% B over 157 min, to 30% B over 30 min, to 60% B over 10 min, and finally to 90% over 1 min. The data was acquired using a top20 method in positive data-dependent acquisition mode using a Nanospray Flex™ ion source ES072, at spray voltage 2000 V, and ion transfer tube temperature 275 °C. The mass range was set to *m/z* 350-1600. The MS/MS spectra were generated using a normalised collision energy at 28. The unassigned charge states and charge states 1, 7, or higher were excluded from fragmentation. Dynamic exclusion was set to 20 s. The data was acquired using Thermo EASY-nLC 1200: 3.2.0 SP2 (version 4.1.4.1), Orbitrap Exploris 480 (tune software version 3.1.279.9) and Xcalibur (version 4.6.67.17). For protein identification, MaxQuant software (Version 2.0.3.0; Max Planck Institute of Biochemistry) and a decoy human UniProt database (2022-01) were used. The variable modifications oxidation (M), N-terminal acetylation, deamidation (N, Q) and the fixed modification carbamidomethyl cysteine were considered in the search. Peptides and proteins, which were identified with a false discovery rate of 1% were used for quantification. For quantification, match between runs and label-free quantitation (LFQ) algorithms were used. Downstream analysis was done in R (Version 4.0.4). Protein groups were filtered by reverse hits, proteins only identified by site, potential contaminants and for valid values in at least 60% of the samples. LFQ values were log2 transformed and missing values were replaced by random values from a normal distribution with a width of 0.3 and a downshift of 1.8. For statistical testing moderated t-test (limma package[57]) and two-sided t-test (rstatix[58]) with Benjamini–Hochberg correction for multiple comparisons were used.

### Reporting summary

Further information on research design is available in the Nature Portfolio Reporting Summary linked to this article.

## Data availability

The glycomics and proteomics raw data generated in this study have been deposited to the ProteomeXchange Consortium via the PRIDE repository under the dataset identifier PXD050530. The processed glycomics data generated in this study, including *O*-glycan structures, CCSs and MS/MS spectra, have been deposited in the UniCarb-DR database under accession codes: 533 (PGM); 534 (healthy sputum); 535 (CF sputum). The processed glycomics data generated in this study, including the annotated MS/MS spectra of *O*-glycan structures identified, are provided in the Supplementary Data 1. The processed glycomics data generated in this study, including the identified *O*-glycan structures with CCS, putative structure, area ratios from ₚGCLC-TWIMS and TIMS data are provided in the Supplementary Data 2. The processed proteomics data generated in this study, including protein intensities reported by the MaxQuant identification software, are provided in the Supplementary Data 3. Source data are provided with this paper. The source data underlying Fig. 6 are provided as a Source Data file. Source data are provided with this paper.

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

## Acknowledgements

We thank the Deutsche Forschungsgemeinschaft (DFG, German Research Foundation) for financing this work under: the CRC 1449 "Dynamic Hydrogels at Biointerfaces" – Project ID 431232613: L.B., K.F., M.S., S.Y.G., M.A.M., P.M. and K.P. –, the CRC 1340 "Matrix in Vision" – Project ID 372486779: Ł.P. and K.P.–, and G.M.V. and K.P. are grateful for the funding by the European Union's Horizon 2020 Research and Innovation Programme grant number 899687—HS-SEQ.

## Author contributions

*O*-glycomics: K.P. directed the research. L.B. and K.P. designed the research. L.B., N.G.K. and K.P. supervised the study. L.B., G.R.E. and M.S. conducted the sample preparation. L.B., G.R.E., M.S., Ł.P. and M.G. contributed to the acquisition of experimental data. L.B., C.J. and N.G.K. analyzed data and interpreted the results. L.B., C.J., G.R.E., M.S., L.P., M.G., G.M.V., W.B.S., N.G.K. and K.P. contributed to the redaction of the manuscript. Proteomics: P.M. directed and supervised the research. K.F. conducted the sample preparation and the acquisition of experimental data. K.F. analysed data, and L.B. and K.F. interpreted the results. L.B., K.F. and P.M. contributed to the redaction of the manuscript. Sputum collection: M.A.M. directed the sputum and ethical approval collection. S.Y.G. and M.A.M. supervised and conducted the sputum and ethical approval collection. L.B., S.Y.G. and M.A.M. contributed to the redaction of the manuscript.

## Funding

## Competing interests

The authors declare no competing interests.
