## [Peer Review File · Nature Communications]

Reviewers' Comments:

Reviewer #1:

Remarks to the Author:

In the manuscript by Bechtella et al., the authors investigate the use of trapped ion mobility spectrometry (TIMS) to separate released O-glycans prior to mass spectrometry (MS) analysis. They show that the TIMS platform effectively separates the O-glycans and allows for the identification of many different structures from porcine gastric mucins and outperforms traditional LC and other IMS modalities like TWIMS. Overall this is a very well-written article and clearly demonstrates the advantages of TIMS for glycomic analysis. I have very few comments and suggestions:

- I believe Figure 4a is in the wrong place in the text, and Figure 4c is never referenced in the text.
- It would be worthwhile for the authors investigate whether the use of TIMS leads to higher levels of in-source fragmentation.
- In Figure 6, what fold change was used for the cutoff? It seems like the settings are very lenient in determining statistical changes between the samples.
- "Overall, the data shows that the mucin fraction collected by size exclusion chromatography contains mostly MUC5AC and MUC5B." This data was not included here, please add an SI figure to address this.
- Some of the figures (e.g., 7c) have gray boxes present

Reviewer #2:

Remarks to the Author:

The authors present the O-glycan profile of clinical samples of healthy vs. Cystic fibrosis sputum obtained by TIM-MS/MS ion mobility. The results are compared, and found consistent, with LC-MS studies which are currently considered as the gold standard for glycomics. Firstly, this work is remarkable with regards to the methodology. To my knowledge, this is the first IMS study of O-glycans from clinical samples and the authors demonstrate that the structural information generally obtained by LC-MS analysis in a couple of hours can be obtained in only minutes with superior repeatability; which may open the way to a radical change of practice in clinical glycomics. Secondly, the authors show an additional level of structural characterisation in the resolution of isomeric structures (Fig 8), which becomes possible using high-resolution IMS and offers new insights in the description of glycosylation deviations/anomalies in pathological context.

The work is extremely carefully conducted with a thorough initial benchmarking study on commercially available porcine gastric mucins for which reference LC-MS and lower resolution IM-MS data (TWIMS) are available (ref 23). The authors have reproduced LC and TWIMS data for comparison with TIMS and their data support the superiority of the proposed approach. On one hand the TIMS separation competes with the LC separation at a reduced time and increased repeatability; on the other hand the TIMS resolution is significantly higher than the TIMS resolution (with equivalent speed).

At first, one might find that the methodology of this work is only incremental because O-glycans profiling by IMS was reported by others in 2019 with the lower separative power available at the time. I would like to stress that the (arguably incremental) improvement in IMS resolution yields entirely original and non-incremental results because a critical threshold was crossed: previously hypothetical structures are now resolved and fully characterised by additional MS/MS analysis. CCS data are reported in accessible databases, which certainly magnifies the impact of this work and will ease the adoption of this high-throughput glycomics workflow by others.

A first "real world" demonstration is presented in the second part of the manuscript, where the authors build on the benchmark study and reference CCS data to compare O-glycosylation in healthy and CF patients. The results are convincing and further supported by comparison with LC data reported by others, in particular regarding the loss of glycan core diversity and the increase of sialylation in CF patients.

I expect that this work will constitute an important milestone in the development of IMS-based high-throughput clinical glycomics.

Suggested changes:

- Typo ref 23 (capital O)
- The comment concerning the CCS of linear vs. branched structures (section CCS based structural characterisation) is confusing because references to Figure 3 and ref 23 are blended in the same sentence, and more importantly possibly irrelevant in this work. While this is a general trend, the results of work curiously deviates from the rule. The structures Fig 3 m/z 749 and fig 8 m/z 587 make significant exceptions and should be discussed.
- The numbers used in mobilograms to represent peak areas look like the number used for the core structures, this is too confusing. Could you use percentages for the areas maybe ?
- Fig 7: sulfated groups are mentioned in the caption but not shown on the structures. Could it be added ?
- There is one point of methodology that remains unclear to me in the sputum section. In the gastric mucin studies TIMS are sufficient for separation and the structures are confirmed with MS/MS, then reference CCS are recorded for future analysis. This suggest that MS/MS will not be necessary in the future, yet sputum data are presented in TIM MS/MS mode. Why not in TIM-MS ? Is there a threshold of complexity/heterogeneity which will limit the applicability of TIM-MS ?

Reviewer #3:

Remarks to the Author:

The authors describe the use of trapped ion mobility spectrometry to characterize various O-glycans. This work was performed carefully and is likely of interest to the readership of Nature Communications. I recommend minor comments to be addressed as shown below.

1) Sentence: "The collision cross section (CCS) measured from mobility experiments is a molecular property and corresponds to the area of the ion that collides with the mobility gas."

CCS is only measured in DTIMS and can be calculated through calibration in TIMS and TWIMS. This sentence should be rephrased. Also, the authors should mention that CCS is the rotationally averaged ion neutral collision cross section.

2) Sentence: "Unlike LC retention times, the CCS of an ion is instrument independent and can be calculated theoretically."

This is confusing. LC times can certainly also be instrument and column dependent. Also, I would argue that theoretical methods have not caught up to the resolution that cIMS and TIMS can provide.

3) Sentence: "The resolving power of IMS is mostly governed by the dimensions of the drift cell."

This is true for TWIMS platforms (e.g., cyclic and SLIM), but not true for TIMS used in this work.

4) The authors should explicitly mention during the results and discussion portion that the TWIMS used is a 25 cm device, whereas the commercially available cyclic IMS (using TW) has much higher resolution.

5) I assume TIMS has higher resolution and could give at least 1 decimal place for the CCS values?

6) In some of the mobilograms, multiple peaks are present for 1 compound. Since these are reduced, what is the origin of multiple peaks? Gas-phase conformations?

7) The dextran ladder used for CCS calibration: were these also reduced? If not, were the anomers observed?

8) My major point of contention is related to the authors using different CCS calibrants in TIMS and TWIMS. It has been previously published that molecular class specific biases can exist depending on calibrant class being different than molecule class. The tune mix is far different in structure than the dextran ladder used, so there will obviously be differences in the CCS values derived between TWIMS and TIMS. I don't see the need for using tune mix at all. Also, the way the mobility calibration section is worded reads confusing.

REVIEWERS' COMMENTS

#Answer to reviewers#

Reviewer #1

In the manuscript by Bechtella et al., the authors investigate the use of trapped ion mobility spectrometry (TIMS) to separate released O-glycans prior to mass spectrometry (MS) analysis. They show that the TIMS platform effectively separates the O-glycans and allows for the identification of many different structures from porcine gastric mucins and outperforms traditional LC and other IMS modalities like TWIMS. Overall this is a very well-written article and clearly demonstrates the advantages of TIMS for glycomic analysis. I have very few comments and suggestions:

#We thank the reviewer for the positive assessment of our manuscript and addressed their comments below.#

- I believe Figure 4a is in the wrong place in the text, and Figure 4c is never referenced in the text.

#We thank the reviewer for this comment. We assume the reviewer is referring to Figure 5 which contains three parts a, b and c and was indeed referenced inconsistently. Figure 5 has been moved further down in the manuscript for more clarity, and now appears in the right paragraph after the title "TIM-MS/MS for glycans structural assignment".#

- It would be worthwhile for the authors investigate whether the use of TIMS leads to higher levels of in-source fragmentation.

#We thank the reviewer for raising this point. The two utilized instruments are indeed different when it comes to ion activation. The LC-MS experiments were performed using an ESI source while the TIMS experiments used an off-line nano-ESI source. As part of a previous study, we qualitatively evaluated the level of activation in both setups (and a third TWIMS instrument with a nESI source) using fragmentation-sensitive glycosaminoglycans. These experiments showed that the TIMS separation is activating the ions to a much lesser extent than the injection and separation of ions in the IM cell of TWIMS instruments. The amount of activation directly in the source (i.e. before the TIMS/TWIMS separation) is rather low and comparable in both cases.#

- In Figure 6, what fold change was used for the cutoff? It seems like the settings are very lenient in determining statistical changes between the samples.

#We thank the reviewer for this comment. In the plot of Figure 6 there is no fold change cut-off applied. Using a fold change cut-off of 1.5 in the representation does not change the interpretation of the data, as the considered mucins, MUC5AC and MUC5B, show \$\log_2(\text{fold change})\$ values of 2.1 (FC=4.3) and -4.2 (FC = 18.4), respectively.

The depiction of the volcano plot is indeed lenient with determining statistical changes between the samples. This is first because the aim of these proteomics experiments is to estimate changes in mucins in these two specific samples, contributing to the glycosylation profiles studied in this work, and not inferring information from overall protein changes. Secondly, statistical changes could not be used to claim differences between populations but only between the two sputum samples used in this work. A different FDR cut-off also does not have any impact on our interpretation as MUC5AC and MUC5B, the main proteins here discussed, are above other commonly used cut-offs.#

- "Overall, the data shows that the mucin fraction collected by size exclusion chromatography contains mostly MUC5AC and MUC5B." This data was not included here, please add an SI figure to address this.

#We thank the reviewer for pointing out the lack of clarity of this statement. The proteomics data were obtained on the whole sputum, without separation by size-exclusion chromatography. As the mucins were collected in the void volume, we assume that we collected mucins with a molecular weight exceeding 500 kDa. To enhance accuracy, the statement has been revised as follows: "Overall, the data shows that the sputum samples contain MUC4, MUC1, MUC16, MUC7, MUC5AC and MUC5B. As size exclusion chromatography with a mass cut-off of 600 kDa was used to collect the mucin fraction before performing glycomics, we assume that the sputum samples used for glycomics contain mostly MUC5AC and MUC5B."#

- Some of the figures (e.g., 7c) have gray boxes present

#We thank the reviewer for spotting this. We have provided all revised figures with a white background.#

Reviewer #2

The authors present the O-glycan profile of clinical samples of healthy vs. Cystic fibrosis sputum obtained by TIM-MS/MS ion mobility. The results are compared, and found consistent, with LC-MS studies which are currently considered as the gold standard for glycomics. Firstly, this work is remarkable with regards to the methodology. To my knowledge, this is the first IMS study of O-glycans from clinical samples and the authors demonstrate that the structural information generally obtained by LC-MS analysis in a couple of hours can be obtained in only minutes with superior repeatability; which may open the way to a radical change of practice in clinical glycomics. Secondly, the authors show an additional level of structural characterisation in the resolution of isomeric structures (Fig 8), which becomes possible using high-resolution IMS and offers new insights in the description of glycosylation deviations/anomalies in pathological context.

The work is extremely carefully conducted with a thorough initial benchmarking study on commercially available porcine gastric mucins for which reference LC-MS and lower resolution IM-MS data (TWIMS) are available (ref 23). The authors have reproduced LC and TWIMS data for comparison with TIMS and their data support the superiority of the proposed approach. On one hand the TIMS separation competes with the LC separation at a reduced time and increased repeatability; on the other hand the TIMS resolution is significantly higher than the LC resolution (with equivalent speed).

At first, one might find that the methodology of this work is only incremental because O-glycan profiling by IMS was reported by others in 2019 with the lower separative power available at the time. I would like to stress that the (arguably incremental) improvement in IMS resolution yields entirely original and non-incremental results because a critical threshold was crossed: previously hypothetical structures are now resolved and fully characterised by additional MS/MS analysis. CCs data are reported in accessible databases, which certainly magnifies the impact of this work and will ease the adoption of this high-throughput glycomics workflow by others.

A first "real world" demonstration is presented in the second part of the manuscript, where the authors build on the benchmark study and reference CCS data to compare O-glycosylation in

healthy and CF patients. The results are convincing and further supported by comparison with LC data reported by others, in particular regarding the loss of glycan core diversity and the increase of sialylation in CF patients.

I expect that this work will constitute an important milestone in the development of IMS-based high-throughput clinical glycomics.

#We thank the reviewer for the positive assessment of our manuscript and addressed the suggested changes below.#

Suggested changes:

- Typo ref 23 (capital O)

#We thank the reviewer for this comment. This has been corrected in the revised manuscript.#

- The comment concerning the CCs of linear vs. branched structures (section CCS based structural characterisation) is confusing because references to Figure 3 and ref 23 are blended in the same sentence, and more importantly possibly irrelevant in this work. While this is a general trend, the results of work curiously deviates from the rule. The structures Fig 3 m/z 749 and fig 8 m/z 587 make significant exceptions and should be discussed.

#We thank the reviewer for this suggestion. Although branched structures tend to show lower CCS, this is indeed not a strict rule and exceptions are often observed. The statement has therefore been removed from the revised manuscript to avoid any confusion.#

- The numbers used in mobilograms to represent peak areas look like the number used for the core structures, this is too confusing. Could you use percentages for the areas maybe ?

#We thank the reviewer for this very helpful suggestion and have modified Figures 3 and 7 accordingly.#

- Fig 7: sulfated groups are mentioned in the caption but not shown on the structures. Could it be added ?

#We thank the reviewer for this remark. The sulfate groups are denoted in the structures with an S that is attached to the individual building block symbol. Prominent examples are three abundant glycans in healthy sputum, in which the sulfate is attached to the Galactose at the 6-branch(Figure 7a). CF sputum on the other hand shows a low degree of sulfation (Figure 7c). None of the major structures is sulfated (Figure 7a).#

- There is one point of methodology that remains unclear to me in the sputum section. In the gastric mucin studies TIMS are sufficient for separation and the structures are confirmed with MS/MS, then reference CCs are recorded for future analysis. This suggest that MS/MS will not be necessary in the future, yet sputum data are presented in TIM MS/MS mode. Why not in TIM-MS ? Is there a threshold of complexity/heterogeneity which will limit the applicability of TIM-MS ?

#We thank the reviewer for this comment. In our study we present the CCS parameter as a structural information, complementary to tandem MS data, to assign glycan structures in untargeted glycomics experiments. Our results indeed highlight the potential of using CCS values for the structural assignment of glycans in targeted glycomics experiments. In practice however, this will require an extensive data base of CCS values, which for O-glycans does not exist yet. The reported CCSs have been uploaded to Unicarb-DR as a first step, but more CCSs are needed to allow a fully comprehensive targeted analysis by TIM-MS in the future.#

Reviewer #3

The authors describe the use of trapped ion mobility spectrometry to characterize various O-glycans. This work was performed carefully and is likely of interest to the readership of Nature Communications. I recommend minor comments to be addressed as shown below.

#We thank the reviewer for the positive assessment of our manuscript and addressed the comments below.#

1) Sentence: "The collision cross section (CCS) measured from mobility experiments is a molecular property and corresponds to the area of the ion that collides with the mobility gas."

CCS is only measured in DTIMS and can be calculated through calibration in TIMS and TWIMS. This sentence should be rephrased. Also, the authors should mention that CCS is the rotationally averaged ion neutral collision cross section.

#We thank the reviewer for this comment. The sentence has been modified in the revised manuscript by: "The collision cross section (CCS) retrieved from mobility experiments is a molecular property and corresponds to the rotationally averaged area of the ion that collides with the drift gas. Depending on the type of instrument, CCSs can be measured directly or estimated after calibration."#

2) Sentence: "Unlike LC retention times, the CCS of an ion is instrument independent and can be calculated theoretically."

This is confusing. LC times can certainly also be instrument and column dependent. Also, I would argue that theoretical methods have not caught up to the resolution that cIMS and TIMS can provide.

#We thank the reviewer for this comment. The statement has been modified in the revised manuscript as follows: "Unlike P_{GC} LC retention times, the CCS of an ion is instrument independent and can in principle be calculated theoretically."#

3) Sentence: "The resolving power of IMS is mostly governed by the dimensions of the drift cell."

This is true for TWIMS platforms (e.g., cyclic and SLIM), but not true for TIMS used in this work.

#We thank the reviewer for spotting this inconsistency. The sentence has been modified in the revised manuscript as follows: "The resolving power of traditional IMS techniques, such as DTIMS and TWIMS, is mostly governed by the dimensions of the drift cell."#

4) The authors should explicitly mention during the results and discussion portion that the TWIMS used is a 25 cm device, whereas the commercially available cyclic IMS (using TW) has much higher resolution.

#We thank the reviewer for this suggestion. We have included a precision in the revised manuscript: "Recent advances in instrumentation address this issue either by performing cyclic IMS experiments in a closed-loop system,[Giles 2019] or by using distinct cell geometries and electric fields."#

5) I assume TIMS has higher resolution and could give at least 1 decimal place for the CCS values?

#We thank the reviewer for this suggestion. The resolving power of TIMS instruments is indeed remarkable and species with very similar CCS can be resolved. Likewise, they usually exhibit a staggering reproducibility with very little variation between different runs. However, CCS values are affected by many different parameters and procedures, which makes it very challenging to determine them accurately. In the present study, CCSs were estimated after calibration with calibrants (dextran) which themselves have a CCS error above 1%. Showing decimal places would therefore not be justified by the data.#

6) In some of the mobilograms, multiple peaks are present for 1 compound. Since these are reduced, what is the origin of multiple peaks? Gas-phase conformations?

#We thank the reviewer for this comment. We have not identified conformers in our data. All structures identified by MS/MS show a single unique mobility peak in the experiment. The different mobility features observed for one particular m/z value (not necessarily one compound) are isomeric glycans of same mass but different structure, which have been identified by MS/MS.#

7) The dextran ladder used for CCS calibration: were these also reduced? If not, were the anomers observed?

#We thank the reviewer for raising this point. The dextran oligosaccharide ladder used for calibration was not reduced because the calibration has to be performed using exactly the same species as used in the reference.(ref 54)

A separation of α and β anomers was not observed at the utilized conditions. The parameters were optimized to separate a diverse range of glycan structures with large differences in length and size from complex samples. It is possible to resolve anomers using IMS condition optimized for this purpose, as studied for example by Ben Faleh *et al.* on a SLIM IMS device. (<https://doi.org/10.1021/acs.analchem.2c01181>) or Ujma *et al.* on cyclic TWIMS (<https://doi.org/10.1007/s13361-019-02168-9>). However, this was not the focus here.#

8) My major point of contention is related to the authors using different CCS calibrants in TIMS and TWIMS. It has been previously published that molecular class specific biases can exist depending on calibrant class being different than molecule class. The tune mix is far different in structure than the dextran ladder used, so there will obviously be differences in the CCS values derived between TWIMS and TIMS. I don't see the need for using tune mix at all. Also, the way the mobility calibration section is worded reads confusing.

#We fully agree with the reviewer, this is a crucial point. The IMS data acquired on both instruments, TWIMS and TIMS, were calibrated externally using the same batch of dextran as calibrant to estimate CCSs. Calibration of the TIMS $1/K_0$ values with tune mix as suggested by the vendor leads to significant deviation of the estimated CCS from those reported previously.

However, during measurement, the tune mix is required to convert the local transit time into $1/K_0$ values. The statement has been detailed in the manuscript as follows: "As TIMS experiments result in the temporal release of the ions, dependent on the electric field gradient, and not in a drift time, a calibration is needed to convert the ion transit time in the ion mobility cell into a reversed mobility $1/K_0$. This was achieved through the otofControl software (Bruker), using the ESI-Low-concentration tuning mix from Agilent."